# Association between anxiety and non-coding genetic variants of the galanin neuropeptide

Gergely Keszler[1]*, Zsuzsanna Molnár[1], Zsolt Rónai[1], Mária Sasvári-Székely[1], Anna Székely[2], Eszter Kótyuk[2]

1 Department of Medical Chemistry, Molecular Biology and Pathobiochemistry, Semmelweis University, Budapest, Hungary, 2 MTA-ELTE Lendület Adaptation Research Group, Institute of Psychology, ELTE Eötvös Loránd University, Budapest, Hungary

* keszler.gergely@med.semmelweis-univ.hu

## Abstract

### Background

Galanin, an inhibitory neuropeptide and cotransmitter has long been known to co-localize with noradrenaline and serotonin in the central nervous system. Several human studies demonstrated altered galanin expression levels in major depressive disorder and anxiety. Pharmacological modulation of galanin signaling and transgenic strategies provide further proof for the involvement of the galanin system in the pathophysiology of mood disorders. Little is known, however, on the dynamic regulation of galanin expression at the transcriptional level. The aim of the present study was to seek genetic association of non-coding single nucleotide variations in the galanin gene with anxiety and depression.

### Methods

Six single nucleotide polymorphisms (SNP) occurring either in the regulatory 5' or 3' flanking regions or within intronic sequences of the galanin gene have been genotyped with a high-throughput TaqMan OpenArray qPCR system in 526 healthy students (40% males). Depression and anxiety scores were obtained by filling in the Hospital Anxiety and Depression Scale (HADS) questionnaire. Data were analyzed by ANCOVA and Bonferroni correction was applied for multiple testing. Linkage disequilibrium (LD) analysis was used to map two haploblocks in the analyzed region.

### Results and conclusions

A single-locus and a haplotype genetic association proved to be statistically significant. In single-marker analysis, the T allele of the rs1042577 SNP within the 3' untranslated region of the galanin gene associated with greater levels of anxiety (HADS scores were 7.05±4.0 vs 6.15±.15; p = 0.000407). Haplotype analysis revealed an association of the rs948854 C_rs4432027_C allele combination with anxiety [$F(1,1046) = 4.140$, $p = 0.042141$, $\eta 2 = 0.004$, power = 0.529]. Neither of these associations turned out to be gender-specific.

**Data Availability Statement:** The data underlying this study are available on the OSF platform via https://osf.io/4djf6/.

**Funding:** This work was supported by the Hungarian Academy of Sciences project (LP-2018-21/2018), the National Research, Development and Innovation Office Hungarian Scientific Research Funds (K100845, K109549, K124132), and the Hungarian Ministry of Human Capacities ELTE Institutional Excellence Program (783-3/2018/FEKUTSRAT). The funders had no role in study design, data collection and analysis, decision to publish, or preparation of the manuscript.

**Competing interests:** The authors have declared that no competing interests exist.

These promoter polymorphisms are supposed to participate in epigenetic regulation of galanin expression by creating potentially methylatable CpG dinucleotides. The functional importance of the rs1042577_T allele remains to be elucidated.

## Introduction

Common affective disorders such as depression and anxiety emerge against the background of perturbed monoaminergic neurotransmission. Beyond the well-substantiated role of classical neurotransmitters such as norepinephrin and serotonin in the pathogenesis of mood disorders, a number of recent studies implicated coexpressed neuropeptides such as galanin, a 29 amino acid long inhibitory neurotransmitter and trophic factor [1], in anxiety and depression [2]. A Bayesian multivariate analysis of gene factors revealed that the galanin signaling system consisting of the stress-inducible galanin gene and three heptahelical galanin receptors seems to play a seminal role in the development of depression by enhancing the vulnerability to environmental stressors [3].

However, the exact function of galanin in depression still remains elusive. Data gained from animal studies suggest that galanin mediates depression-like behaviour [4], while other authors emphasize its antidepressant-like effects [5, 6].

Experimental data argue that the effects of galanin on anxiety are brain region specific and all three galanin receptors seem to mediate its anxiety-like effects. Intracerebroventricular administration of galanin resulted in anxiolytic-like action in rats [7], while its microinjection in rodent amygdala produced both anxiogenic [8] and anxiolytic effects [9]. The anxiolytic action was more predominant when galanin was infused into the dorsal raphe nucleus in rats [10]. However, intra-dorsal hippocampal administration of galanin induced anxiogenic-like behaviours that could be attenuated with a type 2 galanin receptor antagonist [11].

Importantly, these seemingly contradictory findings could be reconciled by showing that type 1 and type 2 galanin receptors mediate opposite anxiety-like effects in the rat dorsal raphe nucleus [12]. The anxiolytic action of the type 2 galanin receptor is underscored by the fact that Gal2R knockout mice exhibit an anxiogenic-like phenotype [13], while type 3 galanin receptor knockout animals display increased anxiety [14]. Polymorphisms in the gene encoding galanin receptor 2 seem to mediate the effect of environmental stressors and gene-environment interactions [15].

Differential expression of galanin has been implicated in a broad spectrum of neuropsychiatric disorders including post-traumatic stress disorder [16] and Lewy body disorder [17]. Human studies revealed a direct correlation between plasma galanin levels and the severity of major depressive disorder [18]. In contrast, intravenous administration of galanin to depressed individuals exerted fast antidepressant activity [19]. Moreover, profound alterations in mRNA expression and DNA methylation have been found in the galanin system in postmortem brain samples of patients with major depressive disorder [20].

Though the regulation of galanin expression is far from being fully characterized, some transcription factors [21], medicine [22] and exercise [22, 23] have been shown to upregulate galanin transcription. Several lines of compelling evidence suggest that non-coding DNA variations affecting important 5' and 3' regulatory as well as intronic sequences are associated with central nervous system disorders such as panic disorder [24], depression and anxiety [25–28] via altering transcription factor or microRNA binding sites.

As described above, other authors have already demonstrated associations between GAL polymorphisms and mood disorders, but no studies have been done to date to test this possible

association in a non-clinical, largely normal, healthy sample. Thus, our goal was to figure out whether any genetic association is detectable between a set of marker SNPs representing the entire regulatory landscape of the galanin gene and mood characteristics in an ostensibly normal sample consisting of university students with no psychiatric history.

## Materials and methods

### Subjects

To test possible genetic associations between GAL polymorphisms and anxiety and depression, DNA samples from a previous study on anxiety were genotyped for GAL polymorphisms [29]. Thus, participants of the present investigation largely overlap with those of an earlier genetic association study where the associations of glial cell derived neurotrophic factor (GDNF) polymorphisms with mood characteristics were tested [29]. The non-related Caucasian (Hungarian) subjects who participated on a voluntary basis were students from several educational institutions from Budapest, Hungary. They gave written informed consent, provided buccal cell samples and filled out the Hospital Anxiety and Depression Scale (HADS) questionnaire. The study protocol was designed in accordance with guidelines of the Declaration of Helsinki, and was approved by the Scientific and Research Ethics Committee of the Medical Research Council (ETT TUKEB). Inclusion criteria were as follows: no past or present psychiatric history (based on self-report), age between 18–35 years, valid GDNF SNP genotypes and valid self-report data for the HADS scales.

A total of 597 independent subjects were genotyped with a high-throughput system, and filled out the HADS self-report questionnaire. Of them, 589 had no previous psychiatric history and 526 subjects were between 18–35 years. Therefore, we analyzed data from 526 participants (40.7% males, 59.3% females; mean age: 22.01 ± 3.1 years).

### Phenotype characteristics

As described in our previous study on anxiety [29], participants completed the Hungarian version [30] of the Hospital Anxiety and Depression Scale [31]. The HADS questionnaire measures anxiety and depression on 7–7 items, scoring on a 0 to 3 Likert scale. It was originally developed to assess anxiety and depression in case of non-psychiatric hospital patients [31]. However, a great deal of studies have shown that it is an adequate tool to measure mood characteristics on non-clinical samples as well [32]. The Cronbach Alpha values of the HADS scales were adequate (Cronbach Alpha = 0.775 for the anxiety and 0.715 for the depression scale, respectively). The inter-correlation of the two scales was high (r = 0.561, p < 0.001). The sample's mean scores were 6.55 (± 3.7) on the anxiety scale and 2.80 (± 2.7) on the depression scale.

### SNP selection criteria

Single nucleotide polymorphisms (SNPs) were selected for genotyping from the Single Nucleotide Polymorphism database of NCBI (dbSNP) with a minor allele frequency greater than 0.05. The pairwise tagging method using r2 threshold of 0.8 by Haploview was used to determine tagging SNPs based on HapMap data to obtain a proper coverage of the *GAL* gene resulting in a mean distance of 1721 bp between the selected SNPs. Polymorphisms with a reference from previous association studies concerning neuropsychiatric disorders were preferred.

## DNA preparation and SNP genotyping

Genomic DNA samples were isolated from buccal swabs as described previously [29]. Genotypes were determined applying the TaqMan® OpenArray™ Genotyping System (Thermo Fisher Scientific, Waltham, Massachusetts) that is based on sequence-specific, fluorescent Taq-Man probes in combination with a high-throughput PCR system using nanoliter-scale sample volume and post-PCR (endpoint) detection. Genotyping panels were obtained from the manufacturer as immobilized target specific primers and fluorescent probes in a low density array format. Reaction mixtures containing approximately 100 ng DNA (range: 30–150 ng) and the 1× master mix (each deoxyribonucleoside triphosphate and the AmpliTaq Gold DNA-polymerase, provided by the manufacturer) were prepared on a 384-well sample plate and then loaded on the genotyping plates by the OpenArray™ Autoloader. PCR amplification was performed in the GeneAmp® PCR System 9700 (Thermo Fisher Scientific, Waltham, Massachusetts) following the manufacturer's instruction. Endpoint imaging of the allele specific FAM and VIC fluorescent intensities was made by the OpenArray™ NT Imager. Raw data were evaluated by the TaqMan Genotyper v1.2 software.

2% of the DNA samples were repeatedly applied on the OpenArray system, demonstrating a reproducibility exceeding 98%. In addition, a subsample was re-genotyped for two SNPs with a 7300 Real-Time PCR System (Applied Biosystems, Foster City, CA) in triplicates for quality control.

Table 1 summarizes the main characteristics of six SNPs genotyped within the GAL gene. Genotype, chromosomal position, relative distances in the gene body and regional localisation for the tested GAL polymorphisms are presented for each SNP. Genotype frequencies, p values from the Hardy-Weinberg equilibrium (HWE) tests and call rate information are also presented. The rs694066 SNP exhibited an outstandingly small minor genotype frequency (0.8%),

**Table 1. Main characteristics of the studied SNPs in the GAL gene.**

| dbSNP number | | Position | Distance | Region | N | % | HWE | Call rate |
|---|---|---|---|---|---|---|---|---|
| rs948854 | TT | g.68682735 | | 5' flanking | 260 | 50.0 | P = 0.996 | 98.86% |
| | CT | | - | | 216 | 41.5 | | |
| | CC | | | | 44 | 8.5 | | |
| rs2097042 | TT | g.68682846 | | 5' flanking | 233 | 50.7 | P = 0.999 | 87.26% |
| | CT | | 111 | | 188 | 41.0 | | |
| | CC | | | | 38 | 8.3 | | |
| rs4432027 | TT | g.68683779 | | 5' flanking | 251 | 51.3 | P = 0.762 | 92.97% |
| | CT | | 933 | | 194 | 39.7 | | |
| | CC | | | | 44 | 9.0 | | |
| rs694066 | GG | g.68685517 | | intron 2 | 399 | 79.2 | P = 0.682 | 95.82% |
| | AG | | 1738 | | 101 | 20.0 | | |
| | AA | | | | 4 | 0.8 | | |
| rs3136540 | CC | g.68688942 | | intron 5 | 260 | 56.7 | P = 0.938 | 87.26% |
| | CT | | 3425 | | 169 | 36.8 | | |
| | TT | | | | 30 | 6.5 | | |
| rs1042577 | CC | g.68691002 | | 3'-UTR | 180 | 40.1 | P = 0.983 | 85.36% |
| | CT | | 2060 | | 210 | 46.8 | | |
| | TT | | | | 59 | 13.1 | | |

HWE = Hardy-Weinberg equilibrium; UTR = untranslated region. Distance: distance between chromosomal positions of neighboring SNPs in base pairs.

while the others vary between 6.5% and 13.1%. All genotypes were in Hardy-Weinberg equilibrium. Call rates exceeded 85% for each polymorphism.

## Linkage disequilibrium analyses

Lewontin's D' as well as $R^2$ values of linkage disequilibrium were determined using the Haplo-View 4.2 software [33]. Haplotypes were determined by the Phase programme [34–36]. Linkage disequilibrium values of the studied SNPs are displayed in Fig 1. Importantly, both Lewontin's D' and $R^2$ analysis revealed two haplotype blocks. Haplotype block No. 1 encompasses rs948854, rs2097042, rs4432027 and rs694066 SNPs, while rs3136540 and rs1042577 belong to haplotype block No. 2.

## Statistical analysis

Statistical analyses were carried out using SPSS 20.0 for Windows. Chi-square analysis was used to assess reliability of the measured genotype and allele frequencies. Lewontin's D' as well as $R^2$ values of linkage disequilibrium were calculated using the HaploView 4.2 application [33]. Haplotypes were determined by the Phase program. To test gender differences, the Inde-pendent-Samples t-test was used. Correlation analyses were carried out to test the HADS scales relationship with age. One-way analyses of covariance (ANCOVA) was used to test genetic associations of the single and multiple marker analyses in an allele-wise design. Bonferroni correction for multiple testing was used to rule out false positive results [37]. The corrected level of significance was p < 0.00417, as the nominal p (threshold value 0.05) was divided by the number of analyses performed (6 SNPs x 2 HADS scales = 12). Two-way ANOVA was used for testing the effect of prior associations in males and females.

# Results

## Influence of age and gender on anxiety and depression scores

Testing for covariates included testing the relationships between age, gender and the measured phenotypes and genotypes. The HADS anxiety scale showed significant differences across males and females [$t(524) = - 4.054$; $p<0.001$]; namely, females showed a higher mean score

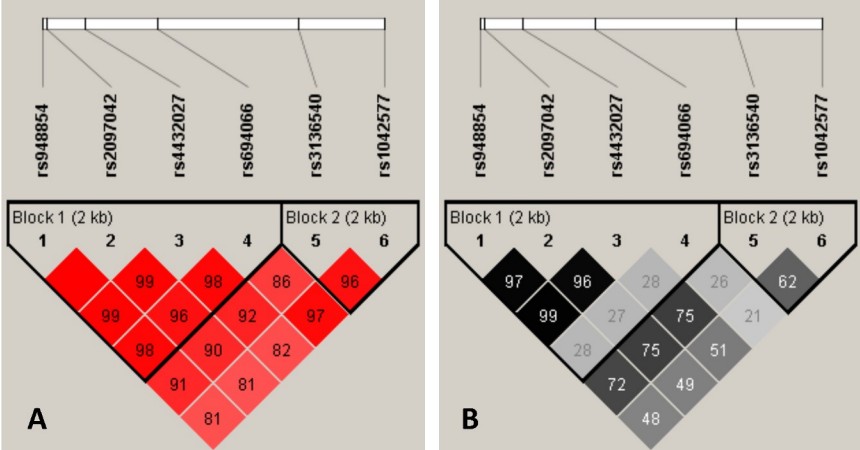

**Fig 1. Linkage disequilibrium plots for the studied GAL SNPs.** Lewontin's D' measure (panel A) and R2 values (panel B) of linkage disequilibrium. Larger figures and darker squares indicate stronger pairwise linkage disequilibrium between two loci.

(7.09) than males (5.78). The HADS depression scale did not differ across genders. For testing the relationship between age and the HADS scales, Pearson correlation was used. The anxiety scale significantly correlated with age ($r$ = - 0.89, $p$ = 0.042), while the depression scale showed no significant correlation. The genotype frequency differences in males and females were tested with Chi-square analyses and no significant differences were observed. The ANOVAs testing the genotype and age association did not show any significant association either. In summary, age and gender were used as covariates in all association analyses on account of their association with the phenotype.

## Single-allele association analyses on mood characteristics with GAL polymorphisms

Table 2 summarizes the results obtained from the single-allele association analyses. Minor allele frequencies, anxiety and depression mean scores for the alleles of each tested SNPs are presented with the corresponding $p$ values. Based on the one-way ANCOVAs, three nominally significant associations were found with the anxiety scale, and one with the depression scale. The results showed higher anxiety mean scores associated with the minor allele of rs948854 [$F(1,136) = 3.865$, $p = 0.049572$, $\eta2 = 0.004$, power = 0.502], rs4432027 [$F(1,974) = 4.349$, $p = 0.037297$, $\eta2 = 0.004$, power = 0.549] and rs1042577 [$F(1,894) = 12.594$, $p = 0.000407$, $\eta2 = 0.014$, power = 0.944]. The minor allele of rs1042577 associated with a higher mean score on the depression scale as well [$F(1,894) = 5.718$, $p = 0.016996$, $\eta2 = 0.006$, power = 0.666]. Bonferroni correction was applied to rule out possible false positive effects. After correcting for multiple testing, association of anxiety and rs1042577 remained significant. In the presence of the minor allele (T) of the rs1042577 the mean anxiety score was significantly higher (7.05 ±4.0) as compared to the major allele (G) carriers (6.15±3.5). The effect of this polymorphism explained 1.4% of the variability of anxiety. In this analysis, the covariate effect of gender but not that of age proved to be significant ($p < 0.001$).

**Table 2. Association analyses between GAL polymorphisms and mood characteristics as measured by HADS questionnaire.**

| dbSNP number | | MAF** | mMAF*** | Anxiety | p | | Depression | p |
|---|---|---|---|---|---|---|---|---|
| rs948854 | C | 0.319 | 0.292 | 6.91 (±4.1) | **0.049572** | | 2.95 (±3.0) | 0.315956 |
| | T | | | 6.45 (±3.5) | | | 2.76 (±2.5) | |
| rs2097042 | C | 0.265 | 0.288 | 6.81 (±3.9) | 0.073644 | | 2.91 (±3.0) | 0.424516 |
| | T | | | 6.36 (±3.5) | | | 2.75 (±2.5) | |
| rs4432027 | C | 0.321 | 0.288 | 6.86 (±4.1) | **0.037297** | | 2.93 (±3.0) | 0.318117 |
| | T | | | 6.34 (±3.5) | | | 2.74 (±2.5) | |
| rs694066 | A | 0.136 | 0.109 | 6.58 (±4.4) | 0.925477 | | 2.83 (±2.9) | 0.728726 |
| | G | | | 6.49 (±3.6) | | | 2.76 (±2.6) | |
| rs3136540 | T | 0.262 | 0.250 | 6.77 (±4.0) | 0.188575 | | 2.83 (±2.8) | 0.602540 |
| | C | | | 6.44 (±3.6) | | | 2.71 (±2.5) | |
| rs1042577 | T | 0.373 | 0.358 | 7.05 (±4.0) | **0.000407**\* | | 3.01 (±3.0) | **0.016996** |
| | C | | | 6.15 (±3.5) | | | 2.59 (±2.4) | |

p: level significance (ANOVA with age and sex as covariant).

\*Significant after Bonferroni correction (12 tests: p<0.004167).

\*\*MAF: minor allele frequency (based on data from the 1000 Genome project).

\*\*\*mMAF: measured minor allele frequency. (Standard deviations are shown in parenthesis)

## Haplotype analyses

As a next step, haplotype analyses were performed involving the three SNPs significantly associated with anxiety comparing the combined effects of the risk alleles versus non-risk alleles. To this end, bipartite and tripartite haplotypes were set up by pairing the most significantly associated rs1042577 3' UTR SNP with one or both of the less significantly associated 5' SNPs as shown in Table 3. Upon comparing haplotype frequencies, it is striking that two major haplotypes are represented with outstanding frequency in each category. For example, in case of the rs1042577—rs948854 haplotypes the non-risk-allele combination (rs1042577 C—rs948854 T) and the one containing both risk alleles (rs1042577 T—rs948854 C) occurred most frequently (0.692 and 0.254, respectively, as compared to 0.040 and 0.014 of the C-C and T-T haplotypes). Similarly, in case of the tripartite haplotype a few combinations were very rare (e.g. rs1042577 C, rs948854 C, rs4432027 T: 0.001) or even unrepresented in the present sample (e.g. C–T–C or T–C- T). In case of all three analyses, the two major haploalleles made up about 95% of all haplotypes. The low frequency of some haplotypes is probably due to the high linkage between the SNPs (see Fig 1).

Thus, the haplotype analyses were carried out between the two most frequent haplotypes. One-way ANCOVAs were applied to both mood dimensions with the haploalleles of the significant SNPs from haploblock 1 (rs948854—rs4432027) and from both haploblocks (rs1042577—rs948854, rs1042577—rs4432027, rs1042577- rs948854—rs4432027). Age and gender were also included as covariates. As it can be seen in Table 4, the haplotype with the risk alleles of both 5' UTR SNPs (rs948854_C—rs4432027_C) showed significantly higher anxiety mean score than the haplotype with non-risk alleles [$F(1,1046) = 4.140$, $p = 0.042141$, $\eta2 = 0.004$, power = 0.529]. Unexpectedly, however, any bi- or tripartite haplotype comprising the risk allele of the 3' UTR SNP (rs1042577) which associated significantly with anxiety even after correcting for multiple testing, did not prove to be associated either with anxiety or depression.

**Table 3. Haplotype frequencies in the studied population.**

| Haplotypes* | N | Frequency |
|---|---|---|
| rs1042577 C—rs948854 T | 728 | 0.692 |
| rs1042577 C—rs948854 **C** | 42 | 0.040 |
| rs1042577 **T**—rs948854 T | 15 | 0.014 |
| rs1042577 **T**—rs948854 **C** | 267 | 0.254 |
| rs1042577 C—rs4432027 T | 729 | 0.693 |
| rs1042577 C—rs4432027 **C** | 41 | 0.039 |
| rs1042577 **T**—rs4432027 T | 14 | 0.013 |
| rs1042577 **T**—rs4432027 **C** | 268 | 0.255 |
| rs1042577 C—rs948854 T–rs4432027 T | 728 | 0.692 |
| rs1042577 C—rs948854 **C**–rs4432027 T | 1 | 0.001 |
| rs1042577 C—rs948854 T–rs4432027 **C** | | 0.000 |
| rs1042577 C—rs948854 **C**–rs4432027 **C** | 41 | 0.039 |
| rs1042577 **T**—rs948854 T–rs4432027 T | 14 | 0.013 |
| rs1042577 **T**—rs948854 T–rs4432027 **C** | 1 | 0.001 |
| rs1042577 **T**—rs948854 **C**–rs4432027 T | | 0.000 |
| rs1042577 **T**—rs948854 **C**–rs4432027 **C** | 267 | 0.254 |

*risk alleles in the haplotypes are displayed bold

**Table 4. Haplotype analysis of risk alleles.**

| Haplotypes* | N | Frequency | Anxiety | p | Depression | p |
|---|---|---|---|---|---|---|
| rs1042577 C _rs948854 T | 728 | 0.692 | 6.41 (±3.54) | 0.075066 | 2.76 (±2.52) | 0.260648 |
| rs1042577 **T**_rs948854 **C** | 267 | 0.254 | 6.83 (±4.10) | | 2.99 (±3.11) | |
| rs1042577 C_rs4432027 T | 729 | 0.693 | 6.41 (±3.54) | 0.059088 | 2.75 (±2.52) | 0.226432 |
| rs1042577 **T**_rs4432027 **C** | 268 | 0.255 | 6.85 (±4.11) | | 3.00 (±3.11) | |
| rs948854 T_rs4432027 T | 742 | 1.000 | 6.42 (±3.52) | **0.042141** | 2.74 (±2.51) | 0.309043 |
| rs948854 **C**_rs4432027 **C** | 308 | 0.293 | 6.89 (±4.05) | | 2.93 (±3.02) | |
| rs1042577 C_rs948854 T_rs4432027 T | 728 | 0.692 | 6.41 (±3.54) | 0.075066 | 2.76 (±2.52) | 0.260648 |
| rs1042577 **T**_rs948854 **C**_rs4432027 **C** | 267 | 0.254 | 6.83 (±4.10) | | 2.99 (±3.11) | |

*risk alleles in the haplotypes are displayed bold. (Standard deviations are shown in parenthesis)

## Gender-specific effects

The HADS anxiety scores differed across the genders, and the covariate effect of gender was also significant in the analysis of rs1042577 and anxiety. Thus, we raised the question whether the effect of this single-locus SNP was gender-specific. To address this issue, a two-way ANOVA was carried out to test whether the effect of rs1042577 in the single-locus analyses differed between males and females. Results are presented in Fig 2. The analysis revealed a significant GAL rs1042577 main effect [$F(1,893) = 11.571$, $p = 0.0007$, η2 = 0.013, power = 0.925], and a significant gender main effect [$F(1,893) = 21.253$, $p < 0.001$, η2 = 0.023, power = 0.996]. However, their interaction was not significant. According to these results, the risk effect of the T allele of the GAL rs1042577 SNP for higher anxiety scores was present in both genders. Males with the T allele showed a mean anxiety score of 6.21 (±3.9) and the females' mean anxiety score with this allele was 7.65 (±3.9) as compared to the 5.48 (±3.6) and 6.61 (±3.4) mean scores with the C allele in males and females. The significant gender main effect is in accordance with literary data [38], suggesting that females in general show higher anxiety scores than males.

Furthermore, we also tested whether the association between the rs948854—rs4432027 haplotype and anxiety is gender specific. The two-way ANOVA showed a significant rs948854—rs4432027 haplotype main effect [$F(1, 1046) = 4.449$, $p = 0.035$, η2 = 0.004, power = 0.559], a significant gender main effect [$F(1,893) = 25.603$, $p < 0.001$, η2 = 0.024, power = 0.999], but the interaction between the rs948854—rs4432027 haplotype and gender was not significant. According to the results (Fig 3), the risk haplotype group showed a higher mean anxiety score both in males (6.24 ± 4.2) and females (7.36 ± 3.9), as compared to the non-risk haplotype groups of males (5.57 ± 3.4) and females (6.98 ±3.5).

## Discussion

Galanin has been shown to co-localize with and modulate the release of norepinephrine and serotonin, principal neurotransmitters in depression and anxiety. In contrast to former research aimed to find genetic associations between GAL gene variants and mood characteristics in subjects with clinical anxiety and depression [, the novelty of the present study is that it was performed on a cohort of apparently normal, healthy volunteers. Candidate single nucleotide polymorphisms were carefully selected to cover 5' and 3' regulatory as well as intronic sequences of the gene, and polymorphisms with previous association data were preferred. Genetic association analyses of six GAL single nucleotide polymorphisms with anxiety and depression have been carried out with one-way analyses of covariance in an allele-wise

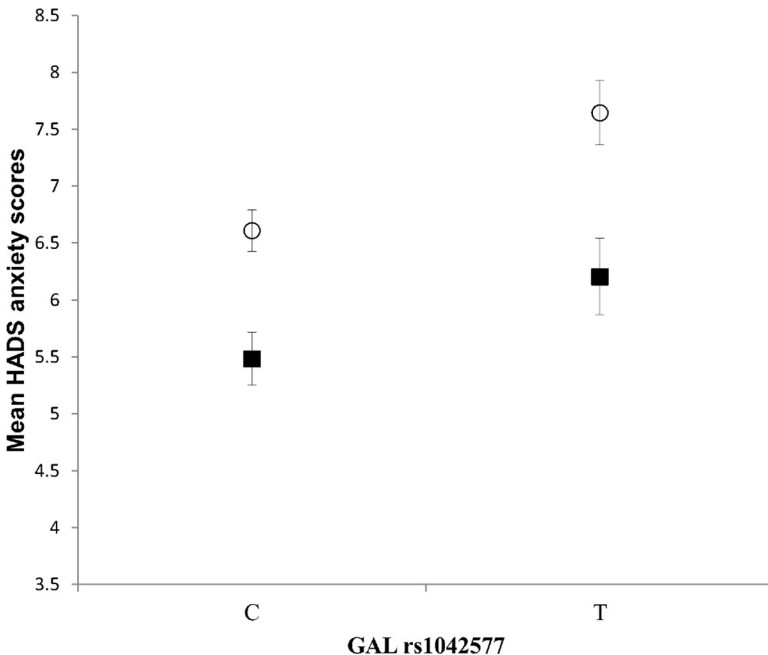

**Fig 2. Effect of GAL rs1042577 alleles on anxiety in males and females.** Mean HADS anxiety scores in males and females as a function of GAL rs1042577 alleles. Open symbols denote females; filled symbols stand for males. Error bars represent standard errors of the mean.

method. Allele frequencies calculated from our population corresponded very well to those available from the 1000 Genomes project. Four nominally significant single-locus associations have been observed in the present study. While GAL rs948854 and rs4432027 SNPs were found to associate with anxiety, the rs1042577 SNP was significantly associated with both anxiety and depression. The association between rs1042577 and anxiety remained significant following correction for multiple testing. The minor allele of this polymorphism is coupled with higher anxiety mean score (T) as compared to the major allele (C). This association was observable in both females and males.

Recent studies conducted on healthy samples also found that HADS anxiety and depression scores varied a lot among university students [39–41]. Importantly, our healthy sample was also quite stratified as 9.9% of the sample (n = 52) scored higher than 11 (mean score of 14.23 ± 2.278) and 11.4% of the sample (n = 60) scored lower than 3 (mean: 1.32 ± 0.673) on the anxiety scale. On the other hand, 9.7% of the sample (n = 51) scored higher than 6 (mean: 9.160 ± 1.902) and 11.6% scored 0 on the depression scale. These data imply that approximately 10% of our admittedly healthy sample had clinical anxiety and depression [32]. It is to note that the frequency of the rs1042577 risk T allele was significantly higher in the top 9.9% than in the bottom 11.4% anxiety groups (50.0% vs. 30.6%, p = 0.005, data not shown).

Haplotype analysis is a very effective tool to test possible allelic interactions. In the present study, comparing the risk allele of rs1042577 from haploblock 2 (see Fig 1) and the risk alleles of the other two SNPs from haploblock 1 (rs948854 and rs4432027) which showed nominally significant associations with anxiety did not yield any significant associations. However, significant association between haplotype rs948854 C_rs4432027_C and anxiety was observed suggesting a possible additive effect of the risk alleles in this haploblock.

The chromosomal distance between these SNPs is slightly more than 1 kb. Importantly, the C allele of each SNP is followed by a guanine nucleotide, creating CpG dinucleotides as

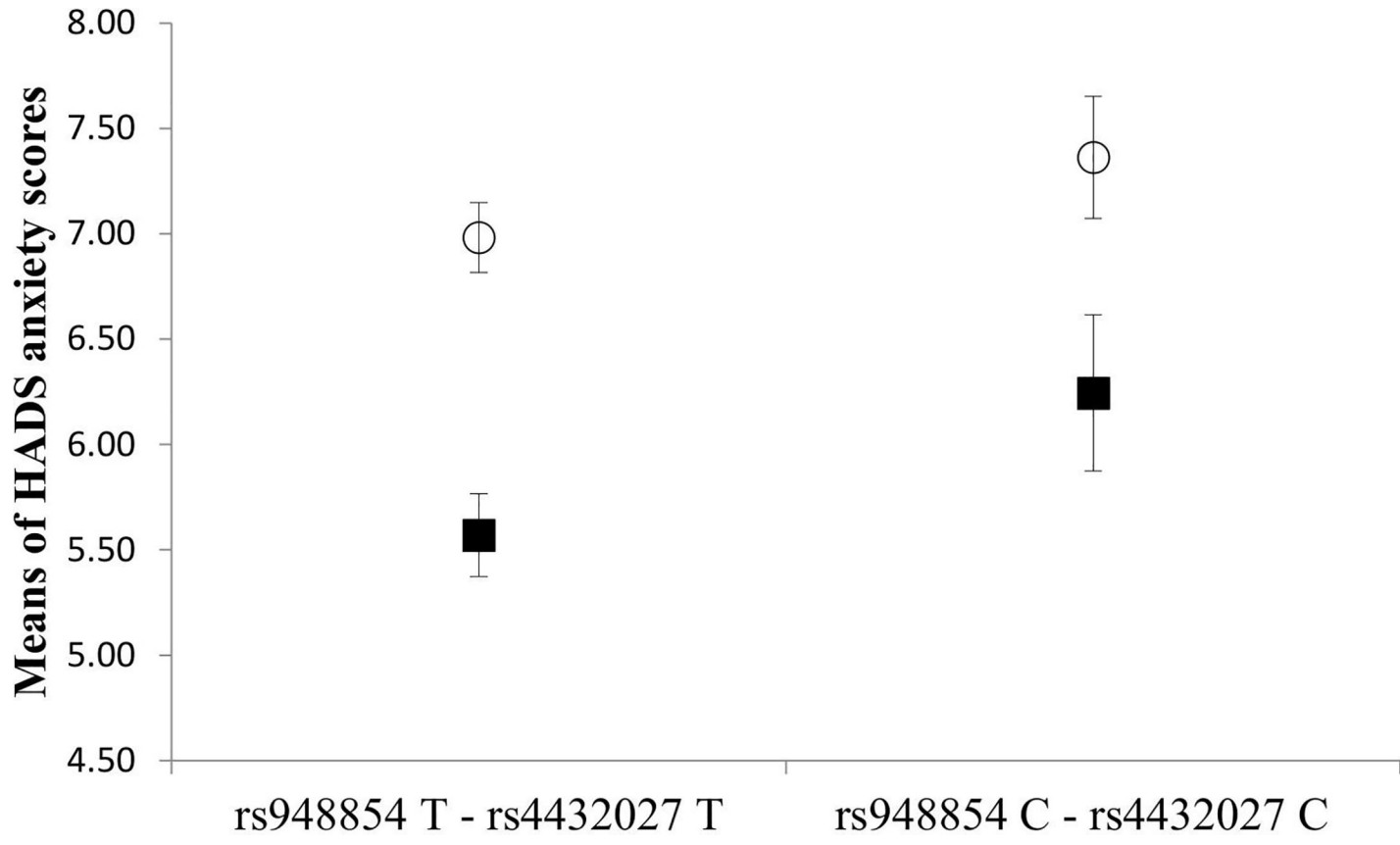

**Fig 3. Effect of the GAL rs948854—rs4432027 haplotype on anxiety in males and females.** Mean HADS anxiety scores in males and females as a function of GAL rs948854—rs4432027 haplotypes. Open symbols denote females; filled ones stand for males. Error bars represent standard errors of the mean.

potential DNA methylation sequences in the promoter [42]. One can therefore assume that their methylation might elicit anxiety via diminished galanin expression. High-resolution methylation mapping of the galanin promoter and functional studies are needed to clarify this assumption. Notably, the role of DNA methylation in the regulation of the galanin system was already suggested by Barde et al. [20].

To our best knowledge, this is the first study in the literature shedding light on the association of the rs1042577 single nucleotide variation with any phenotype. Though this polymorphism was analyzed in a study by Unschuld et al. [24], it was not found to be associated with panic disorder. Apart from psychiatric studies, Schäuble et al. [43] also tested this polymorphism, addressing the role of galanin in fat intake and early onset obesity but failed to find associations.

The rs1042577 SNP is localized to the 3' untranslated region of the galanin gene, a sequence where miRNA binding sites are frequently found. In an attempt to ascribe a functional role to this polymorphism, we searched the online PolymiRTS Database 3.0 (http://compbio.uthsc.edu/miRSNP/) but it turned out that this polymorphism does not affect any known miRNA target sequences. It is also possible that this SNP affects mRNA stability or half-life by allele-specific recruitment of RNA binding proteins but this assumption lacks yet experimental evidence. The rs1042577 SNP is found within a weak CTCF binding site in the 3'UTR. This transcription factor plays multiple roles in transcriptional modulation and chromatin architecture

[44]. The issue whether it really binds the 3' UTR in the context of chromatin needs further investigation.

In summary, in this study we revealed two statistically significant associations between anxiety and the galanin rs948854_C–rs4432027_C haplotype and the rs1042577_T single-locus allele, respectively. These results should be corroborated using clinical samples and functional analyses should also be performed, addressing the role these sequence variants might play in governing galanin expression.

## Author Contributions

**Conceptualization:** Zsolt Rónai, Anna Székely.

**Data curation:** Eszter Kótyuk.

**Funding acquisition:** Anna Székely.

**Methodology:** Zsuzsanna Molnár, Zsolt Rónai.

**Project administration:** Mária Sasvári-Székely, Eszter Kótyuk.

**Software:** Zsolt Rónai.

**Supervision:** Gergely Keszler, Anna Székely.

**Validation:** Mária Sasvári-Székely, Eszter Kótyuk.

**Visualization:** Eszter Kótyuk.

**Writing – original draft:** Gergely Keszler, Eszter Kótyuk.

**Writing – review & editing:** Gergely Keszler.

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
