## [Decision Letter · Decision Letter 0]

2 Sep 2019

PONE-D-19-16922

Association between anxiety and non-coding genetic variants of the galanin neuropeptide

PLOS ONE

Dear Dr Keszler,

Thank you for submitting your manuscript to PLOS ONE. After careful consideration, we feel that it has merit but does not fully meet PLOS ONE’s publication criteria as it currently stands. Therefore, we invite you to submit a revised version of the manuscript that addresses the points raised during the review process.

Be advised that submitting a revision does not guarantee acceptance.

We would appreciate receiving your revised manuscript by Oct 17 2019 11:59PM. To enhance the reproducibility of your results, we recommend that if applicable you deposit your laboratory protocols in protocols.io, where a protocol can be assigned its own identifier (DOI) such that it can be cited independently in the future. For instructions see: http://journals.plos.org/plosone/s/submission-guidelines#loc-laboratory-protocols

We look forward to receiving your revised manuscript.

Kind regards,

Vincenzo De Luca

Academic Editor

PLOS ONE

Journal Requirements:

1. We note that you have stated that you will provide repository information for your data at acceptance. Should your manuscript be accepted for publication, we will hold it until you provide the relevant accession numbers or DOIs necessary to access your data. If you wish to make changes to your Data Availability statement, please describe these changes in your cover letter and we will update your Data Availability statement to reflect the information you provide.

Reviewers' comments:

Reviewer's Responses to Questions

**Comments to the Author**

1. Is the manuscript technically sound, and do the data support the conclusions?

Reviewer #1: Partly

Reviewer #2: Partly

2. Has the statistical analysis been performed appropriately and rigorously? 

Reviewer #1: Yes

Reviewer #2: Yes

3. Have the authors made all data underlying the findings in their manuscript fully available?

Reviewer #1: No

Reviewer #2: Yes

4. Is the manuscript presented in an intelligible fashion and written in standard English?

Reviewer #1: Yes

Reviewer #2: Yes

5. Review Comments to the Author

Reviewer #1: This a study of the association of “six single nucleotide polymorphisms (SNP) occurring either in the regulatory 5’ or 3’ flanking regions or within intronic sequences of the galanin gene” with anxiety and depression scores on the Hospital Anxiety and Depression Scale in 526 healthy students (41% men) ages 18-35 years. As such it is a straightforward correlative study between galanin gene-related SNPs and a psychometric measure of anxiety and depression in normal individuals with no psychiatric illness. A T allele of one SNP and a C-C combination of two other SNPs were associated with greater levels of anxiety in both men and women.

Not being a geneticist, I cannot comment on the methodology used in the genetic analyses; I must assume the analyses were done rigorously and interpreted correctly. The authors did appropriately correct their nominal significance levels for the multiple statistical tests they performed. My concerns are about the nature of the subjects studied vis-à-vis issues of anxiety and depression as psychiatric disorders, to which the authors give detailed attention, and the rating scale used to develop a “phenotype” (subjects with greater or lesser amounts of anxiety and/or depression on the HADS).

Concerning the HADS (ref 31 in the MS), it is a “self-assessment scale…found to be a reliable instrument for detecting states of depression and anxiety in the setting of an hospital medical outpatient clinic. The anxiety and depressive subscales are also valid measures of severity of the emotional disorder… The research was conducted in general medical outpatient clinics on adults of both sexes between the ages of 16 and 65 who suffered from a wide variety of complaints and illnesses.” The HADS thus was not designed to measure anxiety or depression in a sample of normal young adult college students with no complaints or current illnesses. The mean anxiety score (6.55 +/- 3.7 [which I assume is the standard deviation – not stated]) is in the lowest group of scores (Table 1 in ref 31), which consisted of 98% non-and doubtful-cases of clinical anxiety. As well, the mean depression score (2.80 +/- 2.7) was in the lowest group of scores (Table1 in ref 31), which again consisted of 98-99% non-and doubtful-cases of clinical depression. In short, the normal subjects’ scores were clustered near the low end of the HADS, providing little variance for correlational analysis with the genetics data – not surprising for the use of a hospital clinic-derived scale on normal subjects with no complaints or illnesses. More appropriate depression and anxiety scales should have been used to develop the subjects’ “phenotype.”

Given the use of an inappropriate rating scale on normal subjects, the lengthy discussions in the Introduction and Discussion sections of the relationship of galanin to psychiatric illnesses is of little relevance.

I suggest the authors severely shorten the MS and present it as straightforward correlative study between galanin gene-related SNPs and a psychometric measure of anxiety and depression in normal individuals with no psychiatric illness. The authors should provide one or two sentences as rationale for performing the study, limiting any comment about galanin and psychiatric illness to a short, suggestive statement about its relationship to classical neurotransmitters and that it has been found to be altered in certain psychiatric illnesses, and including the list of references.

Reviewer #2: The manuscript "Association between anxiety and non-coding genetic variants of the galanin neuropeptide" by Gergely Keszler and coworkers studies the genetic association of six non-coding single nucleotide variations in the GAL gene with anxiety and depression in healthy young students (mean age 22 years). From their results the authors conclude that only allele T of the rs 10442577 SNP and association of the rs948854C-rs4432027 allele combination are associated with greater lebels of anxiety. Neither association was obseved of other SNP in the GAL gene and depression. The experiments were carried out in an adeguate manner, respect the ethics or publication ethics, and the documentation of the results is satisfactory. Overall, the manuscript is nicely written, satisfies the PLOS ONE criteria for pubblication and is worth publishing. However, I have several comments about the choice of population and the results obtained from this study. The results aren't very interesting. The association between SNP in the GAL genes and anxiety is very low. It is very difficult to demonstrate this association in healthy young people. The works you cited show a strong correlation between SNPs of genes for GAL in anxious subjects, with major depression, panic disorders and multiple schlerosis. As you described in the conclusions, these differences could be ascribed to the fact that in those works they recruit patients with different psychiatric pathologies. You have excluded subjects with psychiatric history from your samples, but it would have been more interesting to study the polymorphisms for the GAL genes in these subjects or in people with family histories of psychiatric illnesses, to try to study the susceptibility of these people for anxiety, depression and other psychiatric disease

6. PLOS authors have the option to publish the peer review history of their article (what does this mean?). If published, this will include your full peer review and any attached files.

Reviewer #1: Yes: Robert T. Rubin, MD, PhD

Reviewer #2: No

---

## [Author Response · Author response to Decision Letter 0]

15 Oct 2019

Response to Reviewers

Fisrt of all, we would like to thank both anonymous Reviewers for thoroughly revising our paper, appreciating its merits and raising valuable concerns which prompted us to rethink several points and to amend the manuscript accordingly. Please find beneath our point-by-point response to your comments and criticism. 

Importantly, a common major concern of both Reviewers was whether it is relevant to study association of GAL polymorphisms with anxiety and depression on a normal, healthy young population rather than on a clinical sample. We admit it should have been better explained in the original manuscript, and are grateful to you for having drawn our attention to our negligence. 

Quotations from reviewers’ comments are written in italics, our responses are typed in normal letters and revised sections of the manuscript are highlighted bold.

Reviewer #2:

1. It is very difficult to demonstrate this association in healthy young people. The works you cited show a strong correlation between SNPs of genes for GAL in anxious subjects, with major depression, panic disorders and multiple schlerosis. As you described in the conclusions, these differences could be ascribed to the fact that in those works they recruit patients with different psychiatric pathologies. You have excluded subjects with psychiatric history from your samples, but it would have been more interesting to study the polymorphisms for the GAL genes in these subjects or in people with family histories of psychiatric illnesses, to try to study the susceptibility of these people for anxiety, depression and other psychiatric disease

As cited in the Introduction of the manuscript, numerous studies found firm genetic association between GAL gene polymorphisms and depression, anxiety, alcohol dependence and panic disorder in psychiatric in- or outpatients. In our study, however, we have intentionally chosen a non-clinical, healthy population to extend the findings of others, that is, to study whether the effect of GAL polymorphisms could be observed on the mood characteristics of a normal cohort. Notably, the anxiogenic effect of a single SNP proved to be significant even in this context, and this effect has never been described before even on a clinical population. This prompts us to check whether the rs1042577 T allele and the rs948854_C_rs4432027_C haplotype as predilection factors are enriched in a clinical sample with anxiety too.

Psychiatric disorders are known to have a strong polygenic genetic background and a lot of environmental stressors (life events) might precipitate the development of symptoms. We chose a young healthy population as the prevalence of negative life events (such as getting divorced, unemployed, widowed, bereaved, having low socio-economic status or afflicted by severe diseases) is far less among these people, making the genetic components more exposed and identifiable. On the other hand, our randomly selected non-clinical population is genetically heterogeneous, i.e. represents a continuous spectrum of mood dimensions, and might comprise subjects with milder (subclinical) forms of anxiety and depression such as GAD (generalized anxiety disorder) or PDD (persistent depressive disorder) as indicated by higher HADS scores on the corresponding scales. Recent studies conducted on healthy samples also found that HADS anxiety and depression scores vary a lot among university students (Andrews et al., 2006; Kebede et al., 2019; Gan and Yuen Ling, 2019). 

To emphasize the above considerations, the bold-marked text above was incorporated in the Discussion and the following paragraph has been inserted in the Introduction:

“As described above, other authors have already demonstrated associations between GAL polymorphisms and mood disorders, but no studies have been done to date to test this possible association in a non-clinical, normal healthy sample. Thus, our goal was to figure out whether any genetic association is detectable between a set of marker SNPs representing the entire regulatory landscape of the galanin gene and mood characteristics on the normal spectrum as well.”

2. The results aren't very interesting. The association between SNP in the GAL genes and anxiety is very low. It is very difficult to demonstrate this association in healthy young people. 

The Reviewer is right in saying that differences between the HADS scores of cohorts with different genotypes are quite low. However, due to the polygenic nature of psychological traits, a single non-coding polymorphism being even in candidate genes typically does not account for greater effects. For comparison, we refer to a paper by de Moor et al. (2012) who performed a meta-analysis on genome-wide association studies for personality traits on a 17,375-strong sample. Despite that large sample size, the variance did not exceed 0.2%. It is of note that in a previous genetic association study (Kótyuk et al., 2013) performed on polymorphisms of the GDNF gene we have got differences commensurable to those in the present manuscript which also proved statistically significant. 

As far as the enrollment of healthy, young people is concerned, we refer to the explanation above. 

Reviewer #1

1. My concerns are about the nature of the subjects studied vis-à-vis issues of anxiety and depression as psychiatric disorders, to which the authors give detailed attention, and the rating scale used to develop a “phenotype” (subjects with greater or lesser amounts of anxiety and/or depression on the HADS).

Concerning the HADS (ref 31 in the MS), it is a “self-assessment scale…found to be a reliable instrument for detecting states of depression and anxiety in the setting of an hospital medical outpatient clinic. The anxiety and depressive subscales are also valid measures of severity of the emotional disorder… The research was conducted in general medical outpatient clinics on adults of both sexes between the ages of 16 and 65 who suffered from a wide variety of complaints and illnesses.” The HADS thus was not designed to measure anxiety or depression in a sample of normal young adult college students with no complaints or current illnesses. 

The Hospital Anxiety and Depression Score (HADS), a psychometric tool was originally developed and has since been widely used to detect depression and anxiety in patients with physical health problems (Zigmond and Snaith, 1983). Over the subsequent decades the test has attained an established role in quantitative psychometrics with almost 5,000 scientific references to date, and it also underwent several re-evaluations and its scope was extended several times. 

By reviewing the literature in 2002, Bjelland et al. concluded that it is an adequate tool to assess mood characteristics in the general population too. One of the founding fathers of the test proposed that the questionnaire was valid when used in community settings too (Snaith, 2003). Finally, the intriguing question whether the HADS could measure anxiety and depression in healthy subjects was addressed by Caci et co-workers (2003). Their landmark study concluded that the HADS inquiry battery is sensitive enough to discriminate between two levels of anxiety in a non-clinical, healthy population. They identified these constructs as ‘anxiety’ and ‘restlessness’ (inner tension), the latter interpreted as low-intensity anxiety with lower scores on the HADS-A scale. Their work has paved the way for dozens of recent analyses deploying HADS to quantitate anxiety and depression in normal healthy subjects, especially in cohorts of university students from all over the world with statistically firm data and conclusions (Andrews et al., 2006; Andrews and Wilding, 2004; Gan and Yuen Ling, 2019; Kebede et al., 2019; Lee et al., 2016; Liu et al., 2009; Moreira de Sousa et al., 2018; Sadeghi et al., 2019; Silva and Figueiredo-Braga, 2018). As it is based on a healthy cohort, our manuscript blends well in this series of investigations, being one of a few recent genetic association studies using HADS scores in healthy populations (Jiménez et al., 2018; Loja-Chango et al., 2016; Kotyuk et al., 2013). The latter is actually a former project of ours that analyzed the same healthy population with HADS to set up associations between non-coding genetic polymorphisms in the GDNF gene and anxiety. Thus, the current manuscript is a direct corollary of that one. 

To briefly reflect the above, the following paragraph has been incorporated to the Phenotype characteristics section of the manuscript: “The HADS questionnaire measures anxiety and depression on 7-7 items, scoring on a 0 to 3 Likert scale. It was originally developed to assess anxiety and depression in case of non-psychiatric hospital patients [31]. However, a great deal of studies have shown that it is an adequate tool to measure mood characteristics on non-clinical samples as well (Caci et al., 2003).”

2. The mean anxiety score (6.55 +/- 3.7 [which I assume is the standard deviation – not stated]) is in the lowest group of scores (Table 1 in ref 31), which consisted of 98% non-and doubtful-cases of clinical anxiety. As well, the mean depression score (2.80 +/- 2.7) was in the lowest group of scores (Table1 in ref 31), which again consisted of 98-99% non-and doubtful-cases of clinical depression. In short, the normal subjects’ scores were clustered near the low end of the HADS, providing little variance for correlational analysis with the genetics data – not surprising for the use of a hospital clinic-derived scale on normal subjects with no complaints or illnesses. More appropriate depression and anxiety scales should have been used to develop the subjects’ “phenotype.”

Yes, in scores like 6.55 � 3.7, the � value denotes the standard deviation. Thank you for your comment, we indicated it in every table of the manuscript.

Well, it is obviously true that the mean scores on the two subscales are near the lower end of the scale in our sample – especially in case of the depression subscale, which is ‘normal’ considering the nature of the data. However, in the context of our sample size and using the same statistical toolbox which most studies did as quoted above – especially ANOVA that focuses on the variability of the dependent variable - , we got statistically significant correlation data even after Bonferroni’s correction for multiple setting. While maximally respecting the opinion of the Reviewer, we take the liberty of hinting that one might not directly compare the HADS scores and cut-off values from the initial study of Zigmond and Snaith conducted on a clinical sample back in 1983 to those measured in our normal, healthy Central European population that differs from theirs in so many respects. 

3. Given the use of an inappropriate rating scale on normal subjects, the lengthy discussions in the Introduction and Discussion sections of the relationship of galanin to psychiatric illnesses is of little relevance. I suggest the authors severely shorten the MS and present it as straightforward correlative study between galanin gene-related SNPs and a psychometric measure of anxiety and depression in normal individuals with no psychiatric illness. The authors should provide one or two sentences as rationale for performing the study, limiting any comment about galanin and psychiatric illness to a short, suggestive statement about its relationship to classical neurotransmitters and that it has been found to be altered in certain psychiatric illnesses, and including the list of references.

As it is a psychiatric genetic study addressing both molecular geneticists, psychologists and psychiatrists, we deemed it important to review and discuss the available literature on galanin and its polymorphisms in sufficient detail to make it understandable for all readers. However, accepting the argumentation of the Reviewer, we have streamlined the Introduction as you suggested regarding correlation between galanin and psychiatric illnesses (see the revised version of the manuscript).

As far as the shortening of the Discussion is concerned, we assume it is important to compare our findings observed in our healthy population to those obtained in transcriptional studies, cell and animal models as well as people with psychiatric problems in order to provide our Readers a comprehensive insight into the relevance of our data in the broader context too. In our opinion, cutting parts of the Discussion would impair the integrity of the paper to a large extent, so we decided not to modify it. 

References 

Andrews B, Hejdenberg J, Wilding J. Student anxiety and depression: comparison of questionnaire and interview assessments. J Affect Disord. 2006 Oct;95(1-3):29-34. Epub 2006 Jun 9.

Andrews B, Wilding JM. The relation of depression and anxiety to life-stress and achievement in students. Br J Psychol. 2004 Nov;95(Pt 4):509-21.

Bjelland I, Dahl AA, Haug TT, Neckelmann D. The validity of the Hospital Anxiety and Depression Scale. An updated literature review. J Psychosom Res. 2002 Feb;52(2):69-77.

Caci H, Baylé FJ, Mattei V, Dossios C, Robert P, Boyer P. How does the Hospital and Anxiety and Depression Scale measure anxiety and depression in healthy subjects? Psychiatry Res. 2003 May 1;118(1):89-99.

de Moor MH, Costa PT, Terracciano A, Krueger RF, de Geus EJ, Toshiko T, Penninx BW, Esko T, Madden PA, Derringer J, Amin N, Willemsen G, Hottenga JJ, Distel MA, Uda M, Sanna S, Spinhoven P, Hartman CA, Sullivan P, Realo A, Allik J, Heath AC, Pergadia ML, Agrawal A, Lin P, Grucza R, Nutile T, Ciullo M, Rujescu D, Giegling I, Konte B, Widen E, Cousminer DL, Eriksson JG, Palotie A, Peltonen L, Luciano M, Tenesa A, Davies G, Lopez LM, Hansell NK, Medland SE, Ferrucci L, Schlessinger D, Montgomery GW, Wright MJ, Aulchenko YS, Janssens AC, Oostra BA, Metspalu A, Abecasis GR, Deary IJ, Räikkönen K, Bierut LJ, Martin NG, van Duijn CM, Boomsma DI. Meta-analysis of genome-wide association studies for personality. Mol Psychiatry. 2012 Mar;17(3):337-49. doi: 10.1038/mp.2010.128. Epub 2010 Dec 21.

Gan GG, Yuen Ling H. Anxiety, depression and quality of life of medical students in Malaysia. Med J Malaysia. 2019 Feb;74(1):57-61.

Jiménez KM, Pereira-Morales AJ, Adan A, Lopez-Leon S, Forero DA. Depressive symptoms are associated with a functional polymorphism in a miR-433 binding site in the FGF20 gene. Mol Brain. 2018 Sep 21;11(1):53. doi: 10.1186/s13041-018-0397-0.

Kebede MA, Anbessie B, Ayano G. Prevalence and predictors of depression and anxiety among medical students in Addis Ababa, Ethiopia. Int J Ment Health Syst. 2019 May 6;13:30. doi: 10.1186/s13033-019-0287-6. eCollection 2019.

Kotyuk E, Keszler G, Nemeth N, Ronai Z, Sasvari-Szekely M, Szekely A. Glial cell line-derived neurotrophic factor (GDNF) as a novel candidate gene of anxiety. PLoS One. 2013 Dec 6;8(12):e80613. doi: 10.1371/journal.pone.0080613. eCollection 2013.

Lee SJ, Park CS, Kim BJ, Lee CS, Cha B, Lee YJ, Soh M, Park JA, Young PS, Song EH. Association between morningness and resilience in Korean college students. Chronobiol Int. 2016;33(10):1391-1399. Epub 2016 Aug 30.

Liu Q, Shono M, Kitamura T. Psychological well-being, depression, and anxiety in Japanese university students. Depress Anxiety. 2009;26(8):E99-105. doi: 10.1002/da.20455.

Loja-Chango R, Pérez-López FR, Simoncini T, Escobar GS, Chedraui P. Increased mood symptoms in postmenopausal women related to the polymorphism rs743572 of the CYP17 A1 gene. Gynecol Endocrinol. 2016 Oct;32(10):827-830. Epub 2016 Apr 27.

Moreira de Sousa J, Moreira CA, Telles-Correia D. Anxiety, Depression and Academic Performance: A Study Amongst Portuguese Medical Students Versus Non-Medical Students. Acta Med Port. 2018 Sep 28;31(9):454-462. doi: 10.20344/amp.9996. Epub 2018 Sep 28.

Sadeghi O, Keshteli AH, Afshar H, Esmaillzadeh A, Adibi P. Adherence to Mediterranean dietary pattern is inversely associated with depression, anxiety and psychological distress. Nutr Neurosci. 2019 Jun 11:1-12. doi: 10.1080/1028415X.2019.1620425.

Silva RG, Figueiredo-Braga M. Evaluation of the relationships among happiness, stress, anxiety, and depression in pharmacy students. Curr Pharm Teach Learn. 2018 Jul;10(7):903-910. doi: 10.1016/j.cptl.2018.04.002.

Snaith RP. The Hospital Anxiety And Depression Scale. Health Qual Life Outcomes. 2003 Aug 1;1:29.

Zigmond AS, Snaith RP. The hospital anxiety and depression scale. Acta Psychiatr Scand. 1983 Jun;67(6):361-70.

---

## [Decision Letter · Decision Letter 1]

24 Oct 2019

PONE-D-19-16922R1

Association between anxiety and non-coding genetic variants of the galanin neuropeptide

PLOS ONE

Dear Dr Keszler,

Thank you for submitting your manuscript to PLOS ONE. After careful consideration, we feel that it has merit but does not fully meet PLOS ONE’s publication criteria as it currently stands. Therefore, we invite you to submit a revised version of the manuscript that addresses the points raised during the review process.

Please, be advised that submitting a revision does not guarantee acceptance.

We would appreciate receiving your revised manuscript by Dec 08 2019 11:59PM. To enhance the reproducibility of your results, we recommend that if applicable you deposit your laboratory protocols in protocols.io, where a protocol can be assigned its own identifier (DOI) such that it can be cited independently in the future. For instructions see: http://journals.plos.org/plosone/s/submission-guidelines#loc-laboratory-protocols

We look forward to receiving your revised manuscript.

Kind regards,

Vincenzo De Luca

Academic Editor

PLOS ONE

Reviewers' comments:

Reviewer's Responses to Questions

**Comments to the Author**

1. If the authors have adequately addressed your comments raised in a previous round of review and you feel that this manuscript is now acceptable for publication, you may indicate that here to bypass the “Comments to the Author” section, enter your conflict of interest statement in the “Confidential to Editor” section, and submit your "Accept" recommendation.

Reviewer #1: (No Response)

2. Is the manuscript technically sound, and do the data support the conclusions?

Reviewer #1: Yes

3. Has the statistical analysis been performed appropriately and rigorously? 

Reviewer #1: Yes

4. Have the authors made all data underlying the findings in their manuscript fully available?

Reviewer #1: Yes

5. Is the manuscript presented in an intelligible fashion and written in standard English?

Reviewer #1: Yes

6. Review Comments to the Author

Reviewer #1: Review of PONE-D-19-16922_R1: The authors have considered the comments of the reviewers on their original MS and have addressed them accordingly. With regard to my comments on the original MS, I still have concerns about the use of the HADS in the sample of students recruited for this study. I do not disagree that the HADS can detect anxiety and depressive symptoms in an ostensibly normal subject sample, such as students, because a large sample will have some clinically anxious and depressed individuals included. And some likely were in the 526 healthy students included in this study. But the data indicate that there were relatively few such students, because, as I noted in my earlier review, the mean anxiety and depression scores were quite low, with small standard deviations. Nevertheless, the phenotypic data are derived from the HADS, and that’s the reality of the situation. I will not argue this point any further, except to say that the authors might emphasize this a bit more, focusing on the unique aspects of their study of galanin genotypes in a largely normal subject sample.

I am less sanguine about the authors’ rationale of not changing their Discussion, other than adding a paragraph (p 18, lines 312-321) that is a bit fuzzy in its reasoning. For example, the first sentence (“Psychiatric disorders…”) is irrelevant to the study. The second sentence (“We chose…”) does not make sense to me; choosing a young healthy “population” (really a sample, not a population) having few negative life events in order to make “genetic components more exposed and identifiable” is illogical – genetic components of what? Young age and health? The next sentence is speculative (“On the other hand, our randomly selected non-clinical population [again, sample, not population]…might comprise subjects with milder…forms of anxiety and depression.”). Yes it might, but it also might not. Why not test this speculation by comparing the top 10% of the sample with the bottom 10%, both in HADS scores and in genetic characteristics? While I usually am against converting continuous data to categorical data, because of the large loss of information, it would be of interest to see if the top 10% of the subjects had HADS scores in the clinical range. (One could get a sense of this as well from the mean scores + 2 standard deviations.) Absent further analysis, the sentence remains speculative and conveys no useful information.

As well, the second sentence of the next paragraph (“It is possible that the association…”) is speculative and non-informative. Yes, anything is possible if the data were different. But they are not.

The above comments on the Discussion section can be summarized by my comment about the MS in my original review: I suggest the authors severely shorten the MS. This particularly includes the Discussion, with elimination of the non-informative aspects and those aspects not directly related to this study.

7. PLOS authors have the option to publish the peer review history of their article (what does this mean?). If published, this will include your full peer review and any attached files.

Reviewer #1: Yes: Robert T. Rubin MD, PhD

---

## [Author Response · Author response to Decision Letter 1]

13 Nov 2019

Authors’ responses to Reviewer 1

Dear Professor Rubin,

Thank you very much for your careful re-evaluation of our revised manuscript. We are very grateful for your valuable suggestions on how to substantiate our largely speculative allegations in the Discussion.

1. We totally agree with your remarks on the use of the HADS questionnaire in a seemingly normal sample of students recruited for this study. To emphasize better that our study differs in this respect from other studies using clinical samples for association studies, the following sentence has been inserted into the introductory section of the manuscript: (association was tested) “in an ostensibly normal sample consisting of university students with no psychiatric history”, and it is highlighted again in the second sentence the Discussion chapter: “In contrast to former research aimed to find genetic associations between GAL gene variants and mood characteristics in subjects with clinical anxiety and depression, the novelty of the present study is that it was performed in a cohort of apparently normal, healthy volunteers.”

2. To expel your concerns regarding our speculative assumption on the potentially marked stratification of our (seemingly) normal sample (students without psychiatric history), we tested whether the HADS scores of the top 10% cohorts of students reach the clinical range on the depression and anxiety subscales – as you suggested. 

As described in the ‘Phenotype characteristics’ paragraph of the ‘Materials and methods’ section, the mean anxiety score in the total sample was 6.55 (± 3.7 standard deviation) and the mean depression score was 2.80 (± 2.7). Upon setting up the “top 10%” and “bottom 10%” cohorts, it turned out that 11.4% of the sample (n=60 individuals) scored lower than 3 and 9.9% of the sample (n=52) scored higher than 11 on the anxiety scale. 9.7% of the sample (n=51) scored higher than 6 and 11.6% of students (n=61) scored 0 on the depression scale. 

Descriptive statistics for these cohorts are as follows. Anxiety mean score ± standard deviation for the anxiety bottom 11.4%: 1.32 ± 0.673; and for the top 9.9%: 14.23 ± 2.278. Depression mean score ± standard deviation for the depression bottom 11.6%: 0 (all depression scores were 0 for all members of this cohort); and for the top 9.7%: 9.160 ± 1.902. Hermann et al. (1995) and Caci et al. (2003) found that the cut-off values for “abnormal”, i.e. clinical depression and anxiety were 11 and 9 HADS scores, respectively. These data corroborate our assumption that a significant portion of our healthy, non-clinical sample had clinical anxiety and depression indeed. 

These findings prompted us to perform genetic association analyses on these cohorts as well. Results of these tests can be seen in the table beneath. The Chi square analysis of allele frequencies in the top versus bottom 10% cohorts revealed statistically significant association of the rs1042577 SNP with anxiety and depression. However, the association between GAL rs948854 and rs4432027 with anxiety was not observable in this context, possibly due to low cohort sizes. 

dbSNP 

number Anxiety Depression

 low- anxiety cohort

(n=60) high- anxiety cohort

(n=52) p low- depression cohort

(n=61) High- depression scores

(n=51) p

rs948854 C 31.4% 39.4% 0.209 36.7% 39.2% 0.696

 T 68.6% 60.6% 63.3% 60.8% 

rs2097042 C 29.6% 37.0% 0.272 37.7% 40.2% 0.726

 T 70.4% 63.0% 62.3% 59.8% 

rs4432027 C 30.2% 39.6% 0.151 36.8% 39.1% 0.736

 T 69.8% 60.4% 63.2% 60.9% 

rs694066 A 14.4% 14.7% 0.950 10.2% 15.2% 0.270

 G 85.6% 85.3% 89.8% 84.8% 

rs3136540 T 27.3% 33.7% 0.322 32.1% 31.7% 0.957

 C 72.7% 66.3% 67.9% 68.3% 

rs1042577 T 30.6% 50.0% 0.005 36.8% 51.2% 0.048

 C 69.4% 50.0% 63.2% 48.8% 

We decided to mention these interesting data in the Discussion section to provide further evidence for the association described between the rs1042577 T allele and anxiety from a different approach and to underpin our speculative sentences with statistical data. The Discussion has been shortened and tautened by removing non-relevant or lengthy speculative sentences, according to your suggestions.

Thank you again for your inspiring comments that made us rethink the concept by obtaining further data in support of the biological relevance of GAL polymorphism in anxiety and depression.

Sincerely yours,

Gergely Keszler

corresponding author

---

## [Decision Letter · Decision Letter 2]

22 Nov 2019

Association between anxiety and non-coding genetic variants of the galanin neuropeptide

PONE-D-19-16922R2

Dear Dr. Keszler,

We are pleased to inform you that your manuscript has been judged scientifically suitable for publication and will be formally accepted for publication once it complies with all outstanding technical requirements.

With kind regards,

Vincenzo De Luca

Academic Editor

PLOS ONE

Additional Editor Comments (optional):

Reviewers' comments:

Reviewer's Responses to Questions

**Comments to the Author**

1. If the authors have adequately addressed your comments raised in a previous round of review and you feel that this manuscript is now acceptable for publication, you may indicate that here to bypass the “Comments to the Author” section, enter your conflict of interest statement in the “Confidential to Editor” section, and submit your "Accept" recommendation.

Reviewer #1: All comments have been addressed

2. Is the manuscript technically sound, and do the data support the conclusions?

Reviewer #1: Yes

3. Has the statistical analysis been performed appropriately and rigorously? 

Reviewer #1: Yes

4. Have the authors made all data underlying the findings in their manuscript fully available?

Reviewer #1: Yes

5. Is the manuscript presented in an intelligible fashion and written in standard English?

Reviewer #1: Yes

6. Review Comments to the Author

Reviewer #1: Much better! The authors did the extra analyses that allow the unique elements to stand out now. I'm pleased my comments were helpful.

7. PLOS authors have the option to publish the peer review history of their article (what does this mean?). If published, this will include your full peer review and any attached files.

Reviewer #1: Yes: Robert T. Rubin, MD, PhD

---

## [Editor Report · Acceptance letter]

18 Dec 2019

PONE-D-19-16922R2 

Association between anxiety and non-coding genetic variants of the galanin neuropeptide 

Dear Dr. Keszler:

I am pleased to inform you that your manuscript has been deemed suitable for publication in PLOS ONE. Congratulations! Your manuscript is now with our production department. 

With kind regards,

on behalf of

Dr. Vincenzo De Luca 

Academic Editor

PLOS ONE